# Differentiating Viral from Bacterial Pneumonia in Children: The Diagnostic Role of Lung Ultrasound—A Prospective Observational Study

**DOI:** 10.3390/diagnostics14050480

**Published:** 2024-02-23

**Authors:** Emil Robert Stoicescu, Roxana Iacob, Adrian Cosmin Ilie, Emil Radu Iacob, Septimiu Radu Susa, Laura Andreea Ghenciu, Amalia Constantinescu, Daiana Marina Cocolea, Cristian Oancea, Diana Luminita Manolescu

**Affiliations:** 1Department of Radiology and Medical Imaging, ‘Victor Babes’ University of Medicine and Pharmacy Timisoara, Eftimie Murgu Square No. 2, 300041 Timisoara, Romania; stoicescu.emil@umft.ro (E.R.S.); dmanolescu@umft.ro (D.L.M.); 2Research Center for Pharmaco-Toxicological Evaluations, ‘Victor Babes’ University of Medicine and Pharmacy Timisoara, Eftimie Murgu Square No. 2, 300041 Timisoara, Romania; 3Field of Applied Engineering Sciences, Specialization Statistical Methods and Techniques in Health and Clinical Research, Faculty of Mechanics, ‘Politehnica’ University Timisoara, Mihai Viteazul Boulevard No. 1, 300222 Timisoara, Romania; 4Department of Anatomy and Embriology, ‘Victor Babes’ University of Medicine and Pharmacy Timisoara, Eftimie Murgu Square No. 2, 300041 Timișoara, Romania; 5Ph.D. School, ‘Victor Babes’ University of Medicine and Pharmacy Timisoara, Eftimie Murgu Square No. 2, 300041 Timisoara, Romania; septimiu.susa@umft.ro (S.R.S.); amalia.constantinescu@umft.ro (A.C.); daiana.cocolea@umft.ro (D.M.C.); 6Department III Functional Sciences, Division of Public Health and Management, “Victor Babes” University of Medicine and Pharmacy, 300041 Timisoara, Romania; ilie.adrian@umft.ro; 7Department of Pediatric Surgery, ‘Victor Babes’ University of Medicine and Pharmacy, Eftimie Murgu Square 2, 300041 Timisoara, Romania; radueiacob@umft.ro; 8Department of Functional Sciences, ‘Victor Babes’ University of Medicine and Pharmacy Timisoara, Eftimie Murgu Square No. 2, 300041 Timisoara, Romania; bolintineanu.laura@umft.ro; 9Center for Research and Innovation in Precision Medicine of Respiratory Diseases (CRIPMRD), ‘Victor Babeș’ University of Medicine and Pharmacy, 300041 Timișoara, Romania; oancea@umft.ro; 10Department of Pulmonology, ‘Victor Babes’ University of Medicine and Pharmacy, 300041 Timisoara, Romania

**Keywords:** trans-thoracic ultrasound, lung ultrasound, children, viral, bacterial, COVID-19, SARS-CoV-2, lung involvement

## Abstract

This prospective observational study aimed to investigate the utility of lung ultrasound (LUS) in diagnosing and managing pediatric respiratory infections, specifically focusing on viral, bacterial, and SARS-CoV-2 infections. Conducted over a period of 1 year and 8 months, this research involved 85 pediatric patients (showcasing a median age of 14 months) recruited based on specific criteria, including age, confirmed infection through multiplex PCR tests, and willingness to undergo LUS imaging. This study employed a 12-area scoring system for LUS examinations, utilizing the lung ultrasound score (LUSS) to evaluate lung abnormalities. The PCR examination results reveal diverse respiratory pathogens, with SARS-CoV-2, influenza, and bacterial co-infections being prominent among the cases. As an observational study, this study was not registered in the registry. Distinct LUS patterns associated with different pathogens were identified, showcasing the discriminatory potential of LUS in differentiating between viral and bacterial etiologies. Bacterial infections demonstrated more severe lung involvement, evident in significantly higher LUSS values compared with viral cases (*p* < 0.0001). The specific abnormalities found in bacterial superinfection can be integrated into diagnostic and management protocols for pediatric respiratory infections. Overall, this research contributes valuable insights into optimizing LUS as a diagnostic tool in pediatric pneumonia, facilitating more informed and tailored healthcare decisions.

## 1. Introduction

Lower respiratory tract infections (LRTIs) are common airway illnesses that affect millions of children worldwide each year and are the most common infections managed in general practice [1,2,3]. Accurate and timely diagnosis is crucial for the effective management and treatment of these diseases [1,4]. While chest radiography (CXR) and computed tomography (CT) have traditionally been used to diagnose pneumonia, they have several limitations, including exposure to ionizing radiation and the need for specialized equipment and trained personnel [5,6,7]. In recent years, lung ultrasound (LUS) has emerged as a promising diagnostic tool for pediatric pneumonia, offering several advantages over traditional methods [5,6,8].

LUS is a non-invasive, radiation-free test that is easy to learn and portable, making it a more feasible option for diagnosing pneumonia in children [8,9]. It involves the use of high-frequency sound waves to create images of the lungs, allowing healthcare professionals to visualize the presence of fluid or inflammation in the lung tissue [10,11]. LUS has been shown to be a sufficiently accurate technique for diagnosing pneumonia in the pediatric population, with a high sensitivity [5,10,12].

In addition to its accuracy in diagnosing pneumonia, LUS has also been shown to be useful in differentiating between bacterial and viral pneumonia in children [13,14,15]. This is important because the appropriate treatment for pneumonia depends on the underlying cause, with bacterial pneumonia typically requiring antibiotics and viral pneumonia requiring supportive care [4,16]. When combined with clinical presentations and laboratory findings, LUS appears to be a promising tool not only in the diagnosis of pediatric pneumonia but also in identifying the causative organism [14,17,18]. Certain features, like having multiple consolidations and interstitial edema, go with viral or atypical bacterial pneumonia compared with bacterial disease, which usually presents as an isolated large consolidation [17,19,20,21].

The use of LUS in pediatric pneumonia has several advantages over traditional methods such as CXR or computed tomography [5,21]. Firstly, LUS is a non-ionizing test, meaning it does not use radiation like CXR [8,22]. This is particularly important in pediatric patients, as they are more sensitive to the effects of radiation [23,24]. Secondly, LUS is portable and can be performed at the bedside, making it more convenient for both patients and healthcare professionals [25,26]. Thirdly, LUS is less expensive than CXR, making it a more cost-effective option for diagnosing pneumonia [27,28].

Despite its advantages, LUS is not without its limitations [9]. It requires trained personnel to perform and interpret the images, and it may not be as accurate in certain cases, such as when the lung fields are clear or when there is a significant overlying pleural effusion [9,29]. Additionally, LUS may not be able to differentiate between viral and atypical bacterial pneumonia, as both may present with similar findings on ultrasound.

This article aims to explore the usefulness of LUS in diagnosing and managing a bacterial or viral etiology of acute lower respiratory tract infections in children. By understanding the role of LUS in diagnosing and managing LRTIs, healthcare professionals can make more informed decisions about the appropriate treatment and care for their patients. We also intend to investigate whether there are differences in LUS findings between different types of viral infections.

We hypothesize that lung ultrasound can effectively distinguish between viral and bacterial causes of acute lower respiratory tract infections in children. Bacterial infections are expected to show more severe lung involvement, with higher LUSSs and a greater presence of specific abnormalities, like confluent B-lines, pleural abnormalities, and subpleural consolidations, compared with viral infections.

## 2. Materials and Methods

### 2.1. Study Design

This study employed a prospective observational design to investigate the lung ultrasonography findings in patients with viral, bacterial, and SARS-CoV2 infection. This research was conducted over a period of 1 year and 8 months, from February 2022 to October 2023, at the Clinic of Infectious Diseases II and the intensive care unit at the ‘Dr. Victor Babes’ Clinical Hospital of Infectious Diseases and Pneumophthisiology in Timisoara. This study was carried out following approval from the Ethics Committee and after obtaining informed consent from all participants. This study adhered to the principles outlined in the Declaration of Helsinki and ensured the de-identification of patient data for analysis and publication. 

Prior to initiating the analysis, a detailed and comprehensive analysis plan was developed. The analysis plan served as a roadmap, outlining the objectives, data selection criteria, methodology, and procedures to be followed throughout the analysis process. This plan was carefully crafted to ensure the integrity, rigor, and reliability of our findings.

As an observational study, this study was not registered in the registry.

### 2.2. Participant Selection

This study recruited children in a sequential manner from the Clinic of Infectious Diseases II and the intensive care unit based on specific criteria, including an age range higher than one month, confirmation of viral, bacterial, or SARS-CoV2 infection with a multiplex PCR (polymerase chain reaction) test, and willingness to undergo lung ultrasound imaging and blood sample collection. Hence, a definitive confirmation of the diagnosis using a multiplex PCR test was necessary for an individual to be included in this study.

The inclusion criteria were based on the presence of LRTIs. We define LRTIs as those individuals that need immediate attention, have acute respiratory symptoms (cough, dyspnea, or respiratory insufficiency), a fever above 38 °C, and show clinical or radiological indicators of lung infiltrates. All subjects underwent testing with a real-time multiplex PCR test, which detects various viruses including adenovirus, coronavirus 229E, HKU1, NL63, OC43, human metapneumovirus, human rhinovirus/enterovirus, human respiroviruses 1, 2, 3, and 4, respiratory syncytial virus, Bordetella parapertussis, Bordetella pertussis, Chlamydia pneumoniae, Mycoplasma pneumoniae, influenza A and B, MERS-CoV, and SARS-CoV-2. The exclusion criteria comprised hospitalized children diagnosed with viral, bacterial, or SARS-CoV-2 infection for less than two days, children with pre-existing chronic lung conditions, such as bronchopulmonary dysplasia, cystic fibrosis, immunodeficiency, and comparable disorders, and children lacking parental or legal guardian consent. These criteria were established to ensure the homogeneity of the study population and the relevance of the findings to the specific research question. 

All the information regarding the process of enrollment, allocation, and analysis of the patients is presented in Figure 1.

### 2.3. Lung Ultrasound Examination

The ultrasonography examinations were conducted without prior knowledge of the infection’s etiology or the outcome of the multiplex PCR test.

Skilled and certified radiologists with over ten years of experience performed lung ultrasound examinations using specific ultrasound machines, settings, and probes. The ultrasound machines used were a portable General Electric Vivid IQ, equipped with a linear probe (9L-RS [2.4–10.0 MHz]) and a convex probe (4C-RS [1.5–5.0 MHz]), and a Philips EPIQ 5, equipped with an L12-5 linear array probe ([12–5 MHz]). 

The lung presetting protocol provided by the manufacturer was used, and the exams were improved according to the patient’s needs. The focus was directed toward the pleural line to achieve a clear visualization of the hyperechoic line, and the exams were concentrated on specified lung regions, adhering to established protocols. This setting can be seen in the attached figures to demonstrate the LUS findings in the Results Section [30].

The lung ultrasound examinations were performed on children admitted to the hospital using a 12-area scoring system, which assigned a score ranging from 0 to 3 points based on the observation of artifacts and the presence or absence of subpleural consolidation. The lung ultrasound score (LUSS) was used to provide a comprehensive summary and a semiquantitative evaluation of each patient’s lung ultrasound findings.

### 2.4. Lung Ultrasound Protocol

The lung ultrasound examinations were conducted using a 12-area scoring system, similar to the lung ultrasound score (LUSS) outlined by Mongodi et al. for COVID-19-related pneumonia in children [31]. 

This scoring system covered six areas on each side of the chest (two anterior, two lateral, and two posterior), delineated by the nipple line. Within each explored area, a scoring system ranging from 0 to 3 points was applied, based on the observation of artifacts and the presence or absence of subpleural consolidation. 

▪A score of 0 was assigned for a normal or physiological pattern displaying A-lines, along with one or two B-lines per intercostal space. ▪A score of 1 indicated the observation of more than two B-lines (referred to as sparse B-lines) per intercostal space, accompanied by pleural abnormalities, such as irregularities or thickening. ▪A score of 2 was allocated for the presence of coalescent or merging B-lines, a ‘white-lung’ appearance, or small peripheral consolidations smaller than 1 cm. ▪A score of 3 was given for substantial peripheral consolidations wider than 1 cm, regardless of the presence of air bronchograms. 

This LUSS scoring system enabled a detailed and nuanced assessment of lung conditions, providing a comprehensive summary of each patient’s lung ultrasound findings [31].

### 2.5. Data Collection and Analysis

The identification data and clinical and ultrasound findings were carefully recorded in a secure computerized database utilizing Microsoft Excel version 2312 ((Build 17126.20132)/9 January 2024).

MedCalc^®^ Statistical Software version 22.017 (MedCalc Software Ltd., Ostend, Belgium; https://www.medcalc.org; accessed on 10 January 2024) was utilized to handle the data and analyses. The Shapiro–Wilk test was used to evaluate the distribution of the plotted data, and non-parametric statistical techniques were used due to a considerable departure from a normal distribution. Central tendency measures were computed using medians and the interquartile range [IQR] for non-parametric variables [32]. The Mann–Whitney U test and cross-tabs were utilized to determine the differences between lung ultrasound findings and LUSSs in children with viral, bacterial, or SARS-CoV2 infections [33].

A sample size calculation was conducted with the following parameters: α = 0.05, β = 0.005, difference in means = 6.5, standard deviation = 3.5, and a viral/bacterial infection ratio of 9:1. The analysis determined that a minimum of 69 individuals were needed for this study, with 62 having viral infections and 7 having bacterial infections. The 9:1 ratio was derived from multiple studies indicating that viruses are responsible for 90% of respiratory infections, while bacteria account for only 10% [34]. After conducting seven statistical tests to evaluate various lung ultrasound findings between patients with viral and bacterial etiologies of infection, a Bonferroni correction was applied to account for multiple comparisons. With a desired significance level of α = 0.05, the corrected value was determined by dividing α by the number of tests conducted, resulting in an adjusted α of approximately 0.0071. This adjusted *p*-value threshold was then used to determine the statistical significance. Any *p*-value below 0.0071 was considered significant after Bonferroni correction, helping to reduce the likelihood of type-I errors in our analyses.

### 2.6. Ethical Considerations

This study followed the criteria stated in the Declaration of Helsinki and was approved by the Ethics Committee of the ‘Dr. Victor Babes’ Clinical Hospital of Infectious Diseases and Pneumophthisiology in Timisoara (number 10289/25 October 2021).

## 3. Results

### 3.1. Baseline Characteristics

Among the 85 patients with acute lower respiratory tract infections, 37 were female, accounting for 43.52% of the total. The median age, measured in months, was 14, with a range of (5, 39). The lowest recorded age was 1 month, while the greatest recorded age was 168 months. A total of 29 individuals, accounting for 34.11% of the total, originated from rural areas. The weight in the center of the distribution was 10.8, while the range between the 25th and 75th percentiles was from 7 to 15. Table 1 displays the basic characteristics of the children involved in this study. The results are compared between the viral cohort and the bacterial cohort. The comparison of features reported as numbers and percentages was conducted using a proportion comparison test, while data expressed as medians and IQRs were analyzed using the Mann–Whitney test.

### 3.2. Viral vs. Bacterial Etiology of Infection

The median lung ultrasound score for all patients was 4, ranging from 1 to 7, with the lowest value being 0 and the highest value being 16. 

Our lot can be divided into two groups based on the cause of infection: a viral etiology and bacterial etiology in cases of co-infection or superinfection. The lot was evaluated by comparing it according to this division. The left posterior inferior region was the most significantly impacted location, with a median value of 1 and an interquartile range of 0 to 1. Furthermore, among the group of individuals with bacterial infections, the left posterior inferior area was shown to be the most impacted, with a median value of 2; (1, 2). Among the individuals with viral infections, the right posterior inferior area was the most impacted, with a median value of 1; (0, 1). 

The analyzed parameters revealed significant differences between viral and bacterial etiologies in terms of LUSSs. Patients with a viral etiology exhibited a median LUSS of 3, ranging from 1 to 6, whereas those with a bacterial etiology displayed a notably higher median LUSS of 10, ranging from 8 to 13.50. The Mann–Whitney test yielded a test statistic Z of −4.27, with a *p*-value of less than 0.0001. Figure 2 displays the LUSS variance between the group with viral infections and those with bacterial infections.

The abnormalities found in the ultrasound examinations are presented and analyzed in Table 2. The LUS findings analyzed were sparse B-lines, confluent B-lines, pleural abnormalities, subpleural consolidation, large consolidation, and pleural effusion.

**Table 2 diagnostics-14-00480-t002:** The lung ultrasound findings analyzed between two groups.

LUS Findings	Number of Patients with Viral Pathologies (*v* = 74)	Number of Patients with Bacterial-Etiology Pathologies (*b* = 11)	Difference	Chi-Squared	Value
Sparse B-lines—Figure 3	55 (74.32%)	11 (100%)	25.68%	3.59	0.0579
Confluent B-lines—Figure 4, Figure 5 and Figure 6	23 (31.08%)	10 (90.91%)	59.83%	14.26	0.0002
Pleural abnormalities—Figure 4 and Figure 5	24 (32.43%)	9 (81.82%)	49.39%	9.72	0.001
Subpleural consolidation of < 1 cm—Figure 6	15 (20.27%)	9 (81.82%)	61.55%	17.69	<0.0001
Large consolidation of > 1 cm—Figure 7	0	5 (45.45%)	45.45%	35.31	<0.0001
Pleural effusion	0	1 (9.09%)	9.09%	6.72	0.009

The ROC curve indicates a sensitivity of 77.14% and a specificity of 80% for viral pneumonia when a LUSS of nine or less is detected. Figure 8 displays the ROC curve for viral infections. The ROC curve shows a sensitivity and specificity of 100% for bacterial pneumonia when large consolidations are found (Figure 9).

## 4. Discussion

This study, comprising 85 patients with acute lower respiratory tract infections, explored the relationship between viral and bacterial etiologies and lung ultrasound findings. Notably, significant differences were observed in baseline characteristics such as age and weight between the viral and bacterial infection cohorts. This suggests varying severity levels between the two groups, in line with the hypothesis. Upon assessing the LUSSs, it became evident that bacterial infections, including superinfections, displayed notably higher median LUSSs compared with viral infections. This supports the hypothesis that bacterial infections are associated with more severe lung involvement, as indicated by the ultrasound findings. Moreover, specific abnormalities such as sparse B-lines, confluent B-lines, pleural abnormalities, and subpleural consolidations were found to be more prevalent in bacterial infections. Large consolidations (> 1 cm) and pleural effusion were exclusively observed in bacterial infections, further bolstering the hypothesis of distinct ultrasound features in bacterial etiologies. The comparison of LUSSs and specific abnormalities between viral and bacterial infections reaffirmed the hypothesis, highlighting significantly higher values in bacterial infections. Additionally, the diagnostic accuracy of lung ultrasound in distinguishing bacterial and viral pneumonia was supported by the ROC curve, which demonstrated good sensitivity and specificity.

In summary, the study findings strongly validate the hypothesis, indicating significant differences in lung ultrasound findings between viral and bacterial etiologies of acute lower respiratory tract infections, particularly in pediatric populations. Bacterial infections exhibit more severe lung involvement and distinct ultrasound features compared with viral infections, underscoring the utility of lung ultrasound in diagnosing and differentiating between these infections.

We consider that one notable strength of this study lies in its comprehensive approach to examining the relationship between lung ultrasound findings and the etiology of lower respiratory tract infections. By including a diverse range of viral and bacterial pathogens, such as SARS-CoV2, influenza, RSV, adenovirus, and various bacterial species, this study captured a broad spectrum of infectious agents commonly encountered in clinical practice. Additionally, the use of multiplex PCR testing provided robust and specific identification of the causative pathogens, enhancing the reliability of the findings. Furthermore, this study employed rigorous statistical analyses, including proportion comparison tests, Mann–Whitney tests, ANOVA, and chi-squared tests, to accurately assess differences in baseline characteristics, lung ultrasound scores, and ultrasound findings between viral and bacterial infections.

The findings of this study are consistent with the existing literature regarding the use of lung ultrasound in distinguishing between viral and bacterial etiologies of acute lower respiratory tract infections [15,35,36]. Previous research has consistently demonstrated that bacterial infections tend to present with more severe lung involvement and distinct ultrasound features compared with viral infections [19,37]. Specifically, the higher median lung ultrasound scores observed in bacterial infections, along with the prevalence of specific abnormalities, such as sparse and confluent B-lines, pleural abnormalities, and subpleural consolidations, align with previous studies highlighting the utility of lung ultrasound in identifying bacterial pneumonia [38]. Moreover, the exclusive presence of large consolidations and pleural effusion in bacterial infections is consistent with the notion that these findings are more commonly associated with bacterial etiologies [35,39]. By reaffirming these trends in a pediatric population and including a diverse range of viral and bacterial pathogens, this study reinforces the existing body of literature on the diagnostic value of lung ultrasound in differentiating between viral and bacterial respiratory infections [40].

The findings also reveal noteworthy trends in the absence of certain LUS abnormalities in viral cases [17,35,41]. Specifically, no patients with viral pathologies exhibited large consolidations of > 1 cm or pleural effusion, which contrasts with the presence of these abnormalities in a significant proportion of bacterial cases [18,35,38]. This absence of certain LUS findings in viral cases could serve as a valuable indicator for clinicians when differentiating between viral and bacterial etiologies, contributing to more accurate and targeted treatment approaches. Overall, these results emphasize the utility of lung ultrasound as a non-invasive and potentially discriminatory tool in the assessment of respiratory pathologies. Furthermore, Table 3 provides an in-depth summary of all the LUS results and might be a significant tool for practitioners in distinguishing between bacterial, viral, and SARS-CoV-2 causes of acute lower respiratory tract infections.

### 4.1. Limitations

The relatively small sample size and single-center design may restrict the generalizability of the findings to broader populations and healthcare settings. The observational nature of this study introduces potential biases, and the operator-dependent nature of lung ultrasound interpretation raises concerns about consistency. Moreover, this study predominantly depends on the results of multiplex PCR tests. Including additional comparable data from alternative imaging techniques or clinical outcomes might strengthen the reliability of the conclusions.

### 4.2. Further Directions

To advance the understanding and integration of lung ultrasound in pediatric respiratory infections, prospective studies with diverse populations are crucial for validation. Standardized protocols for lung ultrasound, coupled with investment in training programs for healthcare professionals, would enhance consistency and proficiency. Additionally, longitudinal studies tracking patients over time could reveal the evolution of lung ultrasound findings and their correlation with clinical outcomes. If validated, integrating lung ultrasound findings into clinical guidelines for pediatric respiratory infections would contribute to comprehensive and standardized diagnostic approaches.

## 5. Conclusions

This study revealed distinct LUS patterns associated with different respiratory pathogens, emphasizing the discriminatory power of this imaging modality. Notably, bacterial infections exhibit more severe lung involvement, as indicated by higher LUSS values and a higher prevalence of specific abnormalities, such as confluent B-lines, pleural abnormalities, and subpleural consolidations. The absence of certain LUS findings in viral cases, such as large consolidations and pleural effusion, stands out as potential indicators for differentiating viral from bacterial infections. 

All in all, this study supports the integration of LUS into the diagnostic and management protocols for pediatric respiratory infections. The nuanced information provided by LUS, coupled with clinical and laboratory data, enhances the ability to differentiate between viral and bacterial etiologies. 

Prospective studies are crucial for validating lung ultrasound in pediatric respiratory infections. Standardized protocols and training programs would enhance consistency. Longitudinal studies could reveal the evolution of ultrasound findings and their clinical correlation. Integration into clinical guidelines would standardize the diagnostic approaches.

## Figures and Tables

**Figure 1 diagnostics-14-00480-f001:**
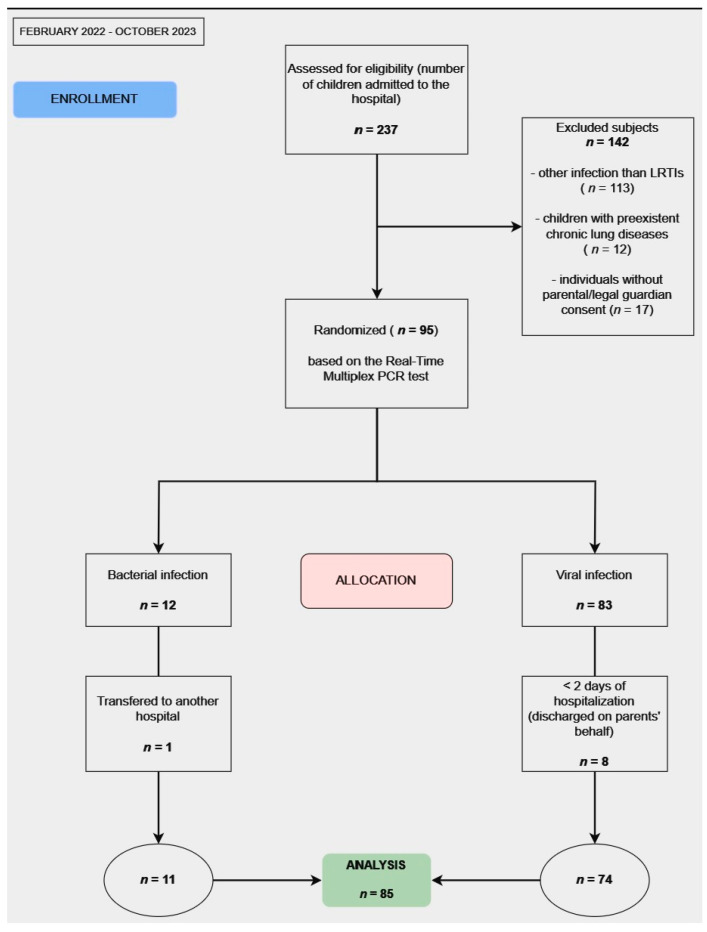
Flow diagram of the progress through the stages (enrollment, allocation, and analysis of the patients).

**Figure 2 diagnostics-14-00480-f002:**
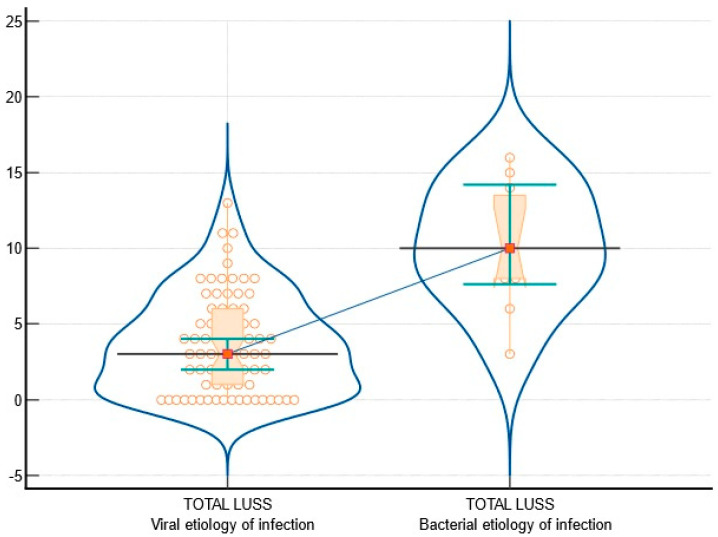
The graphical depiction of LUSSs between the group with viral infections and those with bacterial infections includes notched box-and-whisker and violin plot representations. These plots incorporate dots to represent all data points, as well as horizontal lines, markers, connecting lines, and error bars to indicate 95% confidence intervals for medians.

**Figure 3 diagnostics-14-00480-f003:**
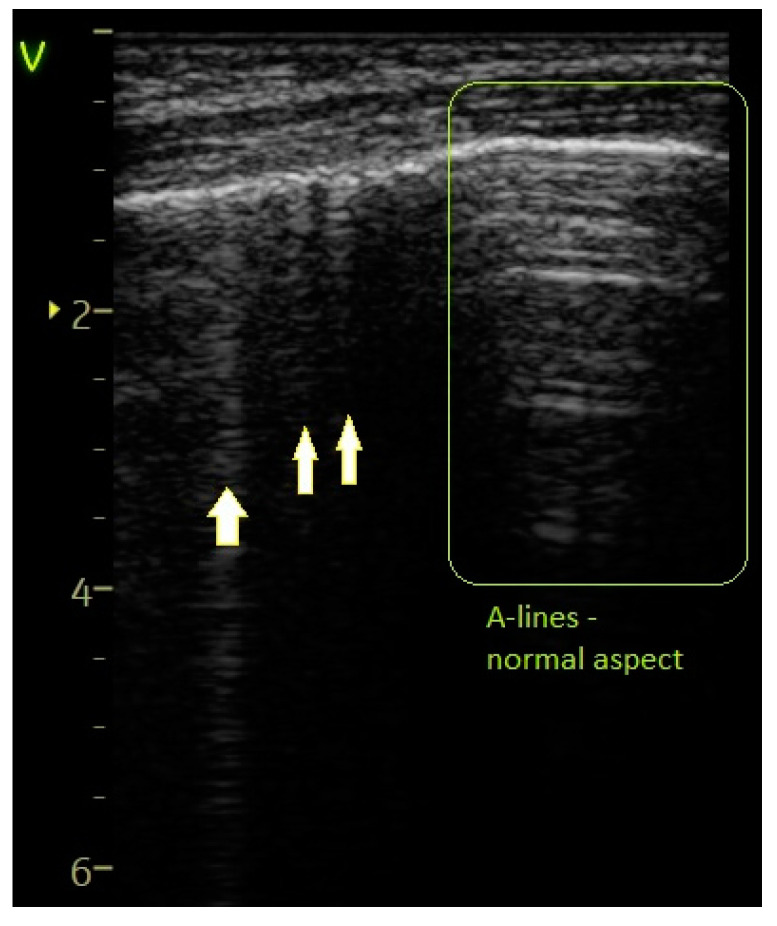
Sparse B-lines (arrows) and A-lines (normal aeration). “V” stands for “vertex” which means the most proximal structure/plan from the transducer.

**Figure 4 diagnostics-14-00480-f004:**
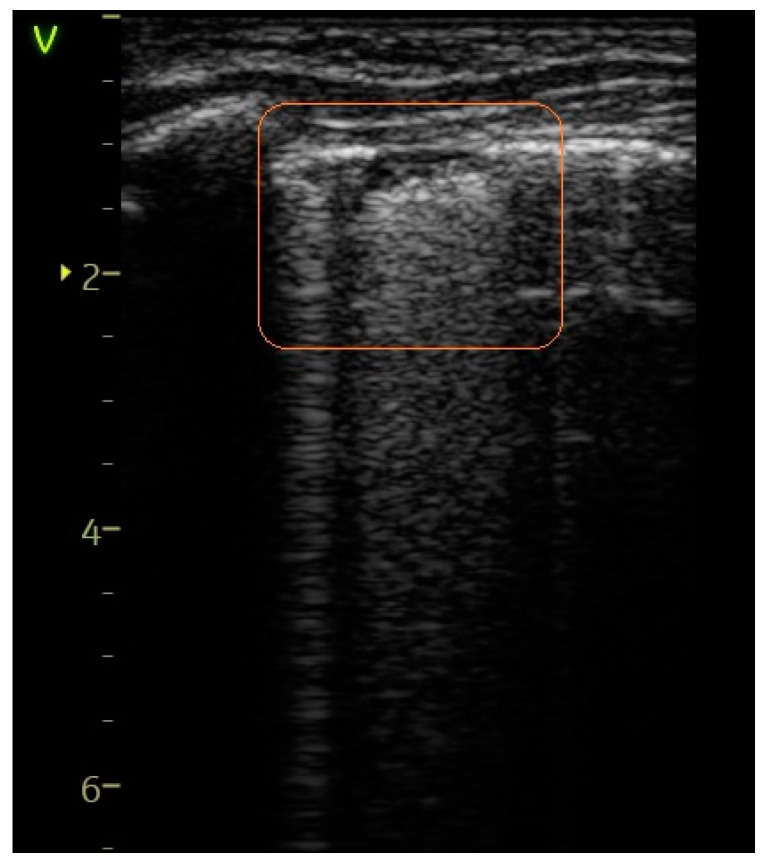
Confluent B-lines and pleural abnormalities are observed in the orange box.

**Figure 5 diagnostics-14-00480-f005:**
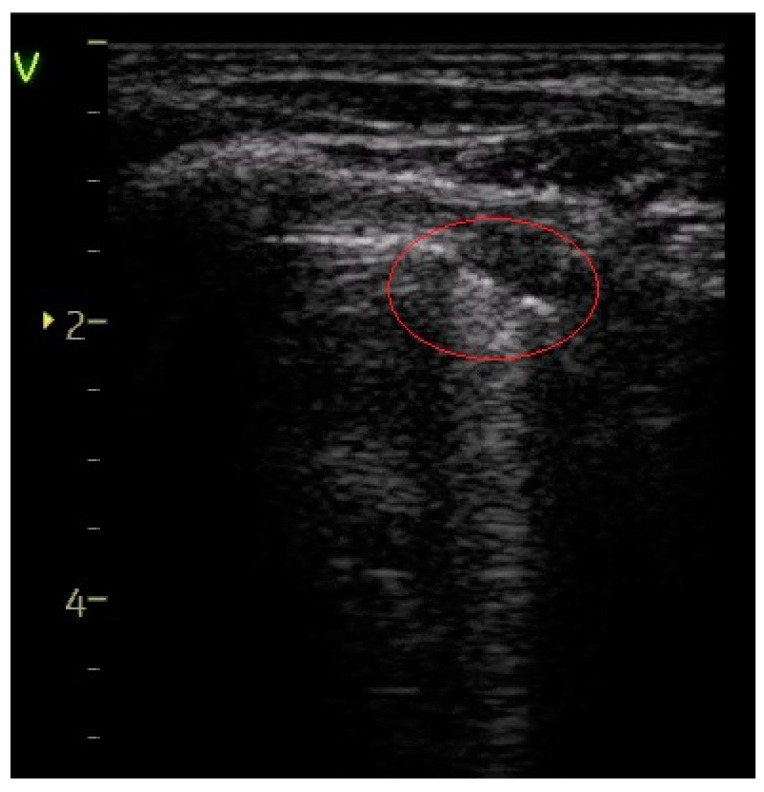
Pleural abnormalities (irregular and thickened) are observed in the red circle.

**Figure 6 diagnostics-14-00480-f006:**
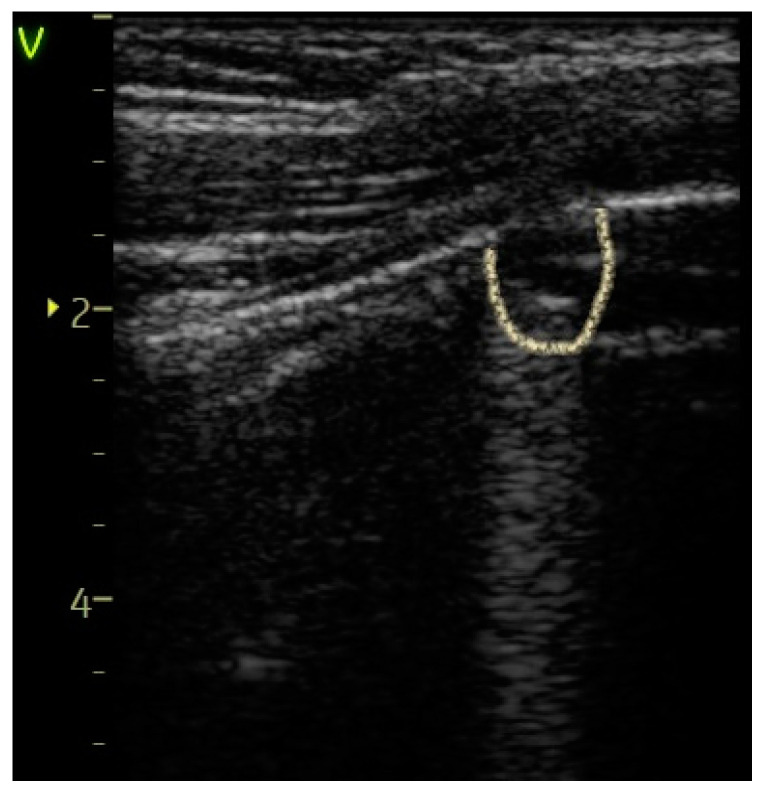
Subpleural small consolidation smaller than 1 cm (area delignated by yellow line) and confluent B-lines.

**Figure 7 diagnostics-14-00480-f007:**
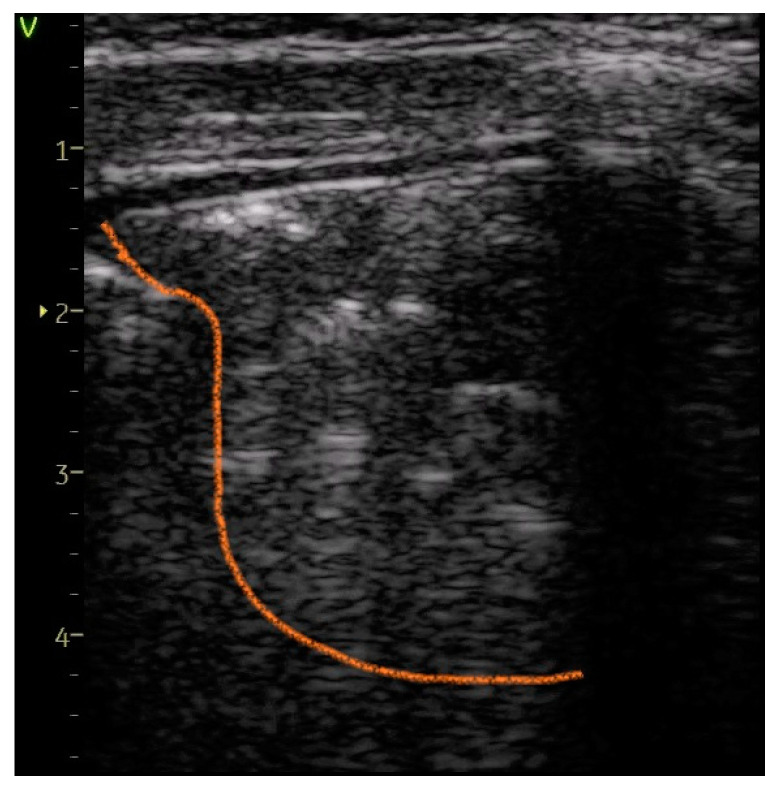
Large consolidation of around 3 cm with air bronchogram (area delignated by orange line).

**Figure 8 diagnostics-14-00480-f008:**
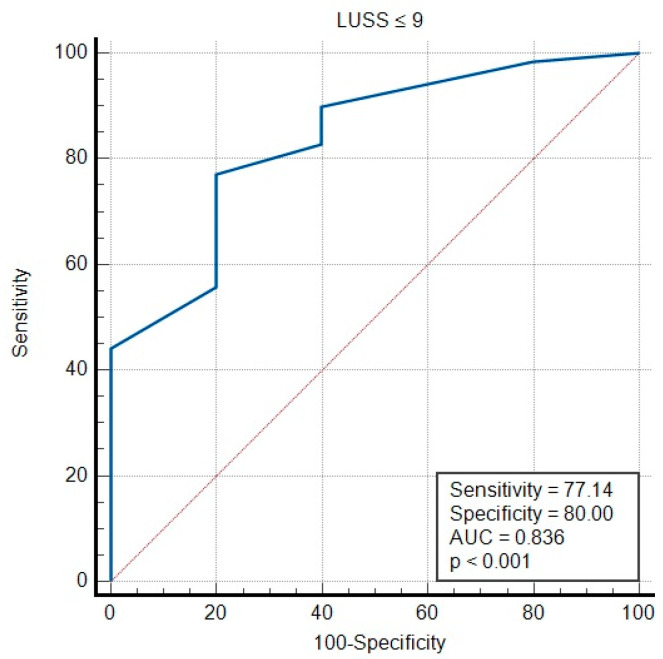
ROC curve for diagnostic accuracy for viral infection.

**Figure 9 diagnostics-14-00480-f009:**
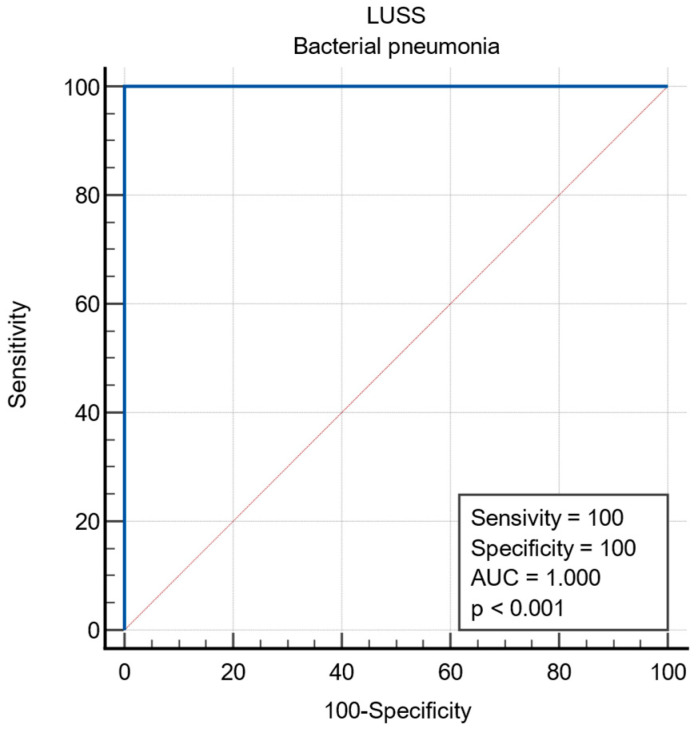
ROC curve for diagnostic accuracy for bacterial infection.

**Table 1 diagnostics-14-00480-t001:** The basic characteristics of the children involved in this study.

Characteristic	Entire Lot (*n* = 85)	Viral Infection (*v* = 74)	Bacterial Infection (*b* = 11)	*p*-Value
Female gender	37 (43.52%)	35 (47.29%)	2 (18.18%)	0.0709
Rural areas	29 (34.11%)	24 (32.43%)	5 (45.45%)	0.5845
Weight (kg)	10.8; (7, 15)	9.8; (7, 15)	14; (10.05, 21.25)	0.0379
Age (months)	14; (5, 39)	12; (4, 24)	36; (23.25, 81.75)	0.0123
Days of hospitalization	5; (4, 7)	5; (4, 6)	7; (6.25, 8.75)	0.0007
Days of convalescence	9; (7, 11.25)	8; (6, 11)	13; (9, 14.75)	0.0070

**Table 3 diagnostics-14-00480-t003:** The summary of lung ultrasound findings.

Finding	Bacterial	Viral	SARS-CoV-2
Distribution of involvement	Lateral and posterior areas	More diffuse	Posterior/lateral subpleural
Pleural line	Irregular near consolidation	Irregular and thickened	Irregular and thickened
Lung parenchyma	B-lines ↓ Focal consolidations with confluent B-lines nearby Air bronchogram	Sparse B-lines ↑ Confluent B-lines ↓	Sparse B-lines Confluent B-lines
Consolidations	Focal consolidations (> 1 cm) and hepatization	< 1 cm	< 1 cm subpleural consolidations
Pleural effusion	+/−	-	-

↑—increased level of. ↓—decreased level of.

## Data Availability

The data are encapsulated within this article. Further details can be obtained upon request from either the primary author or the corresponding author. The data are inaccessible to the public due to the patient privacy regulations governing clinical data.

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
