# Peer review of "Differentiating Viral from Bacterial Pneumonia in Children: The Diagnostic Role of Lung Ultrasound—A Prospective Observational Study"

_diagnostics, 2024, doi:10.3390/diagnostics14050480_

Round 1

Reviewer 1 Report

Comments and Suggestions for Authors

The current study titled “Clarity in Breath: Unveiling Pediatric Respiratory Infections 2 with Lung Ultrasound Precision” Ref: 2868487, deals with an important subject.

LUS is a safe technique that can be adopted as an accessible fast and inexpensive methodology for diagnosis of serious disease.

The study employed small group (85 patients, median age 14 months), with almost fair gender distribution. The net observations due to this clinical studies support the efficiency of the hypothesized methodology for diagnosis of respiratory infection (bacteria/virus). However the limited group considered in this study dimensioned the applicability. Additionally, to assign accuracy of the attend observations, the net assignments should be compared to an approved/usable protocol.

Minor revision is only needed for the addition of a list of abbreviations mentioned throughout the study. The future prespections for enhancing the net observations (that may be in a future separate study) should be highlighted in the conclusion section.

Author Response

Dear reviewer,

Thank you very much for these valuable comments.

We carefully reviewed the text and made necessary corrections to the English language.

We appreciate your ongoing support in enhancing general comprehension of the data presented in the manuscript.

We hope that the changes that have been implemented are to your satisfaction, thereby ensuring that the article fulfills the requisite criteria for publication consideration.

Sincerely,

Authors

Reviewer 2 Report

Comments and Suggestions for Authors

Stoicescu er al performed a prospective study in LUS in children with an acute respiratory infection. Main findings were that LUS patterns differ between patients with a bacterial infection versus patients with a viral infection. I have the following comments.

  1. Is the study registered? If not, why not, and if so, where and please provide this information in the manuscript
  2. Was an analysis plan in place, before starting the analysis?
  3. An hypothesis at the end of the Introduction is missing - please provide the hypothesis tested in the investigation
  4. A sample size calculation is missing, please provide the sample size calculation at the end of Methods, in analysis paragraph
  5. The title is not fitting too well; provide one that covers what you did, and prevent obscure wording
  6. The Introduction is quite long and not spot on - also, there is quite some redundancy — part of this Introduction is actually Discussion
  7. The number of tests you used in high, did you perform correction for that?
  8. A CONSORT could help understand how many and which patients were excluded (assuming you excluded patients, mentioned in the Methods), and then how groups ere formed) - I also missed a table with patient demographics and characteristics
  9. You are splitting the sample in many, maybe too many groups, leading to groups with small numbers, at times even only one patient - I would advise not to do this, create two groups (bacterial, and viral) and compare those - in addition, if you insist, perform a sensitivity analysis wherein you focus on bacterial superinfections
  10. The second part of the analysis, in which you compare the different regions, makes no sense—too many regions with only little observations
  11. If this paper is about diagnostic accuracy, then ROC curves need to be constructed, and you should calculate sensitivity and specificity — this, at best, can be useful in deciding whether based on LUS one could decide not to start antibiotics, for instance
  12. You did not analyze LUS for guidance, remove that part
  13. Discussion is quite long, unfocused and does not stay close to the findings - also, do not repeat findings (in numbers) in Discussion - ideally you start with a summary of findings (in relation to the hypothesis tested), then a paragraph about strengths, then how your findings are in line with existing literature, then how not, then one paragraph on how the information this study brings can be used (in clinical practice, or for future research), and finally the limitations 
Comments on the Quality of English Language

No comments

Author Response

Dear reviewer,

Thank you very much for these valuable comments.

We carefully reviewed the text and made necessary corrections to the English language.

We appreciate your ongoing support in enhancing general comprehension of the data presented in the manuscript.

We hope that the changes that have been implemented are to your satisfaction, thereby ensuring that the article fulfills the requisite criteria for publication consideration.

Sincerely,

Round 2

Reviewer 2 Report

Comments and Suggestions for Authors

PLEASE SEE MY COMMENTS ON YOUR ANSWERS BELOW, WHERE APPLICABLE

1. Is the study registered? If not, why not, and if so, where and please provide this information in the manuscript. 

R: The study is unregistered because it is purely observational and does not require any further engagement in patient management. 

Comment: then please report this in the Abstract and in tekst: ‘this study was not registered in registry.’

2. Was an analysis plan in place, before starting the analysis? 

R: Indeed, we aspired to examine the differences between bacterial and viral infection. Moreover, we aim to determine if there is a predetermined correlation between viral infections based on their etiology. We have attempted to show variations in ultrasonography findings among various types of viral infections.

Comment: this is not what I meant; I meant to ask you if the full analysis plan was in place before you started the analysis

3. An hypothesis at the end of the Introduction is missing - please provide the hypothesis tested in the investigation 

R: Thank you for this valuable comment! We added the hypothesis at the end of the Introduction – ‘Given the emerging utility of lung ultrasound in diagnosing respiratory infections, particularly in pediatric populations, we hypothesize LUS findings will differ significantly between patients with viral and bacterial etiologies of acute lower respiratory tract infections. Specifically, bacterial infections will exhibit more severe lung involvement, including higher LUSS and a higher prevalence of specific abnormalities such as confluent B-lines, pleural abnormalities, and subpleural consolidations, compared to viral infections._’ 

Comment: keep it simple, maybe simply write ‘we hypothesize that LUS findings would be useful to distint patients with viral pneumonia from patients with and bacterial pneumonia.’

4. A sample size calculation is missing, please provide the sample size calculation at the end of Methods, in analysis paragraph 

R: Thank you for this valuable comment! A sample size calculation was conducted with the following parameters: α = 0.05, β = 0.005, difference in means = 6.5, standard deviation = 3.5, and a viral/bacterial infection ratio of 9:1. The analysis determined that a minimum of 69 individuals are needed for the study, with 62 having viral infection and 7 having bacterial infection. The 9:1 ratio was derived from multiple research indicating that viruses are responsible for 90% of respiratory infections, while bacteria account for only 10% [36] 

Coment: teh sampel size calculation should be mentioned in the methods

5. The title is not fitting too well; provide one that covers what you did, and prevent obscure wording 

R: Thank you for your feedback. The title has been modified as per your recommendation. The revised title is 'Utility of Lung Ultrasound in Pediatric Respiratory Infections Differentiation: Insights into Diagnosis and Management - An Observational Study'. We faith that this title is suitable.

Comment: to be honest, I think it is still not what you did, maybe better to write: To diagnostic capacity of LUS to distinct viral from bacterial pneumonia in children’, do not add management, because you did not (even) looked at that

6. The Introduction is quite long and not spot on - also, there is quite some redundancy — part of this Introduction is actually Discussion 

R: Thank you for your feedback. We have reduced the length of the introduction partand we have added the hypothesis at the end of the Introduction. 

7. The number of tests you used in high, did you perform correction for that? 

R: The inclusion criteria were based on the presence of LRTIs. We have defined as LRTI, those individuals that need immediate attention, have acute respiratory symp-toms (cough, dyspnea, or respiratory insufficiency), a fever above 38 °C, and show clinical or radiological indicators of lung infiltrates. All subjects underwent testing with a Real-Time Multiplex PCR Test, which detects various viruses including Adeno-virus, Coronavirus 229E, HKU1, NL63, OC43, Human metapneumovirus, Human rhi-novirus/enterovirus, Humanrespirovirus 1, 2, 3, and 4, Respiratory Syncytial Virus, Bordetella parapertussis, Bordetella pertussis, Chlamydia pneumoniae, Mycoplasma pneumoniae, Influenza A and B, MERS-CoV, and SARS-CoV-2.

Comment: this is not what I meant, the number of statistical test is high, and you need to correct fort his, eg., Bonferoni

8. A CONSORT could help understand how many and which patients were excluded (assuming you excluded patients, mentioned in the Methods), and then how groups ere formed) - I also missed a table with patient demographics and characteristics. 

R: Thank you for this valuable comment. All the information regarding the process of enrollment, allocation and analysis of the patients are presented in the Figure 1 from 2.2 Patients Selection (Figure 1. Flow diagram of the progress through the stages (enrollment, allocation and analysis of the patients). Furthermore, we introduced the Table 2 – The basic characteristics of the children involved in the study. 

9. You are splitting the sample in many, maybe too many groups, leading to groups with small numbers, at times even only one patient - I would advise not to do this, create two groups (bacterial, and viral) and compare those - in addition, if you insist, perform a sensitivity analysis wherein you focus on bacterial superinfections. 

R: You are correct about dividing the group into smaller groups. However, we believe that the sub-chapter 3.2, "Assessment of Lung Ultrasound Score for Viral and Bacterial Etiology," which highlights the differences in ultrasound scores based on the exact etiology determined by PCR test results, is a significant aspect of this article and an original contribution to the field. Chapter 3.2 is the only section that examines the significance of alterations based on the precise cause of the infection. If you strongly believe this method is incorrect, we can eliminate it. 

Comment: yes, I would delete it

10. The second part of the analysis, in which you compare the different regions, makes no sense—too many regions with only little observations. 

R: We consider this strategy advantageous since it allows us to identify areas susceptible to viral or bacterial infections. The table design at the end of the discussions summarizes these centralized data and displays the distribution of LUS changes based on explanation. If you are convinced that this information should be removed, we can erase it. 

Comment: yes, I would delete it

11. If this paper is about diagnostic accuracy, then ROC curves need to be constructed, and you should calculate sensitivity and specificity — this, at best, can be useful in deciding whether based on LUS one could decide not to start antibiotics, for instance 

R: Thank you for your valuable comment. Two ROC curves were presented in the latter part of the results. Unfortunately, the individuals who were enrolled only underwent lung ultrasonography for diagnostic imaging. We have confirmed the infection by the results of the Multiplex PCR test. The ROC curve indicated a sensitivity of 77.14% and a specificity of 80% for viral pneumonia when a LUSS of 9 or less is detected. Figure 9 displays the ROC curve for viral infection. The ROC curve showed a sensitivity and specificity of 100% for bacterial pneumonia when large consolidation are found (figure 10). 

12. You did not analyze LUS for guidance, remove that part 

R: That’s right. This part has been excluded from the discussion. We base the Discussion part on your insightful remarks. 

13. Discussion is quite long, unfocused and does not stay close to the findings - also, do not repeat findings (in numbers) in Discussion - ideally you start with a summary of findings (in relation to the hypothesis tested), then a paragraph about strengths, then how your findings are in line with existing literature, then how not, then one paragraph on how the information this study brings can be used (in clinical practice, or for future research), and finally the limitations 

R: That’s right. Thank you very much. We base the Discussion part on your insightful remarks. The discussion section was completely rewritten.

Author Response

Dear reviewer,

Thank you one more time for these valuable comments.

We carefully reviewed the text and made necessary corrections to the English language.

We appreciate your ongoing support in enhancing general comprehension of the data presented in the manuscript.

We hope that the changes that have been implemented are to your satisfaction, thereby ensuring that the article fulfills the requisite criteria for publication consideration.

Sincerely,

Authors
